Natural disturbance impacts on trade-offs and
co-benefits of forest biodiversity and carbon.
*Proc. R. Soc. B* **288**: 20211631.

ecology, ecosystems, biological applications

biodiversity conservation, carbon sequestration,
carbon storage, climate change, historical
disturbance, primary forest

**Author for correspondence:**
Martin Mikoláš
e-mail: mikolasm@fld.czu.cz

Electronic supplementary material is available
online at https://doi.org/10.6084/m9.figshare.
c.5666862.

# Natural disturbance impacts on trade-offs and co-benefits of forest biodiversity and carbon

Martin Mikoláš[1], Marek Svitok[1,2], Radek Bače[1], Garrett W. Meigs[3],
William S. Keeton[4], Heather Keith[5], Arne Buechling[1], Volodymyr Trotsiuk[1,6],
Daniel Kozák[1], Kurt Bollmann[6], Krešimir Begović[1], Vojtěch Čada[1],
Oleh Chaskovskyy[7], Dheeraj Ralhan[1], Martin Dušátko[1], Matej Ferenčík[1],
Michal Frankovič[1], Rhiannon Gloor[1], Jeňýk Hofmeister[1], Pavel Janda[1],
Ondrej Kameniar[1], Jana Lábusová[1], Linda Majdanová[1], Thomas A. Nagel[8],
Jakob Pavlin[1], Joseph L. Pettit[1,9], Ruffy Rodrigo[1,10],
Catalin-Constantin Roibu[11], Miloš Rydval[1], Francesco M. Sabatini[12,13,14],
Jonathan Schurman[1], Michal Synek[1], Ondřej Vostarek[1], Veronika Zemlerová[1]
and Miroslav Svoboda[1]

[1]Czech University of Life Sciences Prague, Faculty of Forestry and Wood Sciences, Kamýcká 129, Praha 6 Suchdol,
16521 Czech Republic
[2]Department of Biology and General Ecology, Faculty of Ecology and Environmental Sciences, Technical
University in Zvolen, Masaryka 24, Zvolen 96001, Slovakia
[3]Department of Natural Resources, Washington State, 1111 Washington Street SE, Olympia, WA 98504, USA
[4]Rubenstein School of Environment and Natural Resources, University of Vermont, 81 Carrigan Drive, Burlington,
VT, USA
[5]Griffith Climate Change Response Program, Griffith University, Parklands Drive, Southport, Queensland 4222,
Australia
[6]Swiss Federal Institute for Forest, Snow and Landscape Research WSL, Zuercherstrasse 111, Birmensdorf 8903,
Switzerland
[7]Faculty of Forestry, Ukrainian National Forestry University, Gen. Chuprynka 103, Lviv 790 57, Ukraine
[8]Department of Forestry and Renewable Forest Resources, Biotechnical Faculty, University of Ljubljana,
Večna pot 83, Ljubljana 1000, Slovenia
[9]Department of Biology, Minot State University, Minot, ND, USA
[10]Department of Forest Science, Biliran Province State University, Biliran Campus, Biliran 6549, Philippines
[11]Forest Biometrics Laboratory—Faculty of Forestry, 'Stefan cel Mare' University of Suceava, Universitătii Street
no. 13, Suceava 720229, Romania
[12]German Centre for Integrative Biodiversity Research (iDiv) Halle-Jena-Leipzig, Puschstraße 4, Leipzig 04103,
Germany
[13]Martin-Luther University Halle-Wittenberg, Institute of Biology, Am Kirchtor 1, Halle 06108, Germany
[14]Alma Mater Studiorum—University of Bologna, Department of Biological, Geological and Environmental
Sciences, BIOME Laboratory, Via Irnerio 42, 40126 Bologna, Italy

MM, 0000-0002-3637-3074; MS, 0000-0003-2710-8102; GWM, 0000-0001-5942-691X;
HK, 0000-0001-5956-7261; VČ, 0000-0002-3922-2108; DR, 0000-0003-2813-7685;
JP, 0000-0001-8514-3446; MR, 0000-0001-5079-2534; FMS, 0000-0002-7202-7697;
OV, 0000-0002-0657-0114

With accelerating environmental change, understanding forest disturbance
impacts on trade-offs between biodiversity and carbon dynamics is of
high socio-economic importance. Most studies, however, have assessed
immediate or short-term effects of disturbance, while long-term impacts
remain poorly understood. Using a tree-ring-based approach, we analysed
the effect of 250 years of disturbances on present-day biodiversity indicators
and carbon dynamics in primary forests. Disturbance legacies spanning cen-
turies shaped contemporary forest co-benefits and trade-offs, with
contrasting, local-scale effects. Disturbances enhanced carbon sequestration,

reaching maximum rates within a comparatively narrow post-disturbance window (up to 50 years). Concurrently, disturbance diminished aboveground carbon storage, which gradually returned to peak levels over centuries. Temporal patterns in biodiversity potential were bimodal; the first maximum coincided with the short-term post-disturbance carbon sequestration peak, and the second occurred during periods of maximum carbon storage in complex old-growth forest. Despite fluctuating local-scale trade-offs, forest biodiversity and carbon storage remained stable across the broader study region, and our data support a positive relationship between carbon stocks and biodiversity potential. These findings underscore the interdependencies of forest processes, and highlight the necessity of large-scale conservation programmes to effectively promote both biodiversity and long-term carbon storage, particularly given the accelerating global biodiversity and climate crises.

## 1. Introduction

Carbon storage and habitat provisioning for biodiversity are two of the most important ecosystem services provided by forests [1,2]. Forest ecosystems are large terrestrial carbon pools, sequestering approximately 34% of annual anthropogenic carbon emissions [3]. As such, forest management aimed at increasing carbon storage is a major component of natural climate solutions (NCS). Over the next decade, NCS have the potential to cost-effectively provide 37% of carbon mitigation needed to limit global warming to 2°C with a 66% chance [4]. Yet, the effectiveness of carbon storage for climate mitigation depends on long-term forest functionality and integrity, which critically depends on biodiversity [5]. However, abrupt biodiversity declines have been observed in natural forests worldwide, as a result of widespread habitat degradation or fragmentation owing to human impacts on intact forest landscapes [6]. Rapid global climate change and the biodiversity crisis necessitate adaptive policies and strategies, in which forests will play a key role [7,8].

Carbon storage and biodiversity are related to the dynamic nature of forest ecosystems. Disturbance is a primary driver of forest structure, and while disturbance events typically generate the structural variability required to sustain high biodiversity, large pulses of tree mortality and subsequent decomposition can reduce forest carbon stocks [9]. For these reasons, it is extremely difficult to determine whether forests can simultaneously sustain both high carbon and high biodiversity, with previous research from a variety of forest types either demonstrating trade-offs [10] or synergies (i.e. 'co-benefits' [11]).

Recent studies suggest a positive relationship between total carbon storage (i.e. 'stocks') and biodiversity in tropical forests [11,12]. Similarly, there is a positive relationship between carbon stocks and both bird and tree species diversity at landscape scales across Europe and North America [11,13]. Crucially, more detailed stand-scale analyses from temperate regions suggest the opposite pattern [10], and results differ widely depending on the scale of analysis, biogeographic regions or taxonomic groups [10,14–16]. These uncertainties in our understanding of possible trade-offs between forest carbon storage and biodiversity conservation challenge policy development aiming to optimize both objectives, particularly in response to abrupt changes as climate warming alters natural disturbance regimes [9,17,18]. Harnessing the potential of forests to tackle

the climate and biodiversity crises requires improved understanding of natural disturbance processes and their long-term effects on forest carbon dynamics and biodiversity [7,19,20].

Disturbances like wind, insect outbreaks and forest fire can rapidly kill trees over a range of extents, re-shaping forest structure at both stand (e.g. tree age class distribution and seral condition) and landscape (e.g. vegetation pattern and patch mosaics) scales [21]. Because disturbances influence the successional development of recovering vegetation for decades or even centuries, they can have long-lasting effects on forest biodiversity and carbon [19]. Variation in the spatial and temporal scale of disturbances raises many challenges when trying to quantify disturbance effects on forest functions, especially with short-term data. Large-scale studies on the effects of historical disturbances on present day forest functions and biodiversity are rare, however, largely because of the difficulties in reconstructing detailed, long-term histories of natural disturbances and stand development [22]. Only a large-scale and long-term perspective can provide insight on the effects of past disturbances on present-day forest functions [23]. This broad perspective is crucially needed for assessing the vulnerability of forest ecosystems to changing conditions and for developing policy options to simultaneously tackle biodiversity conservation and climate change mitigation [24].

A significant proportion of Earth's forest cover still exists free of direct human intervention in locations known as primary forests (approx. 27%) [25]. Primary forests are the result of complex natural disturbance histories, and are typically highly heterogeneous, both within and among stands that include the range in seral stages as well as old-growth forest [26,27]. Their structural heterogeneity translates into high spatial variability in carbon storage and biodiversity, although primary forests generally maintain high levels of both [28]. Being less influenced by humans compared to managed or secondary forests, primary forests represent natural laboratories for investigating interactions among biodiversity, carbon and disturbance dynamics [29].

Here, we investigated the long-term response of biodiversity indicators (biodiversity potential index and occurrence of an umbrella species, the capercaillie (*Tetrao urogallus* L.)), and forest carbon dynamics (sequestration and total storage) to 250 years of disturbance history across a gradient of disturbance severity and timing. To reconstruct disturbance histories, we collected 7725 tree cores in 30 of the best-preserved primary Norway spruce (*Picea abies* (L.) Karst.) forest stands in temperate Europe. We addressed three main research questions:

(i) how does variation in past disturbance history affect contemporary patterns of biodiversity indicators, carbon storage, and carbon sequestration?

(ii) what is the relative importance of disturbance severity and timing in determining contemporary biodiversity indicators and carbon storage and sequestration? and

(iii) under which disturbance conditions are there co-benefits versus trade-offs between forest biodiversity, carbon storage and sequestration?

## 2. Material and methods

### (a) Study area

We conducted this study in one of the largest remaining contiguous forest ecosystems in Europe—the Carpathian Mountain ecoregion (figure 1), which encompasses the majority of extant

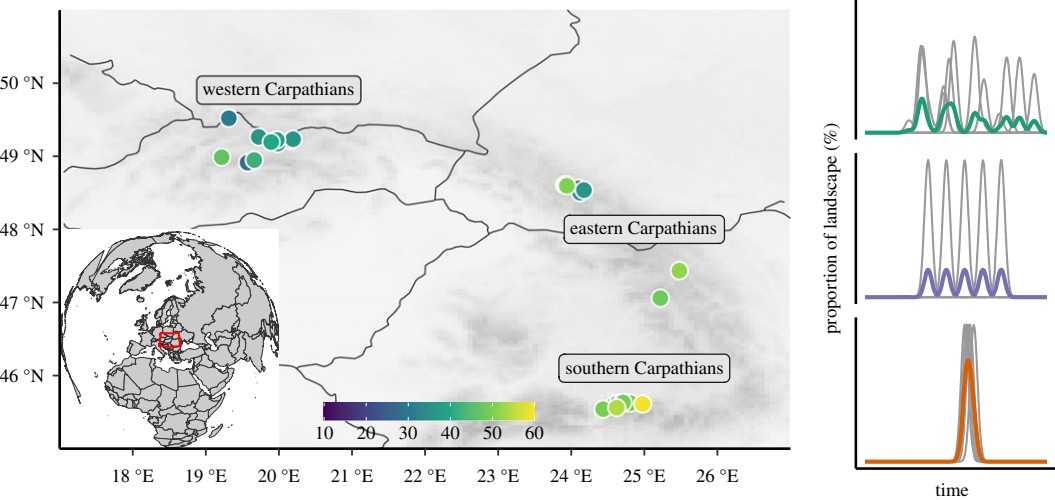

**Figure 1.** Study area and plot locations. Data collection was based on a hierarchical stratified random sampling design. Forest stands (circles) were randomly distributed within remnant primary forest patches and across broad environmental gradients. The colour gradient indicates the maximum severity of historical disturbance of the studied stands. The reconstructed disturbance history for all studied stands is based on the tree ring analyses of 25 trees per plot. Examples of hypothetical disturbance histories (three panels on the right) show moderate (green), low (violet) and high severity (orange) disturbance regimes (the grey line represents the tree level signals, while the coloured line represents the plot-level disturbance signal). The y-axis corresponds to the proportion of forest where a disturbance event caused the removal of the tree canopy, as inferred from tree-rings. (Online version in colour.)

temperate primary spruce forests in Europe [29]. We combined datasets used in previous analyses of disturbance history [21]. The respective field sampling procedures are summarized in the project REMOTE [30]. Thirty primary forest stands with no signs of human management were selected in the subalpine zone of the Carpathian Mountains. Stands with no evidence of direct human influence, such as logging or livestock grazing, were selected with the help of local experts or primary forest inventories [31]. The studied forests occupy altitudes ranging from 1150 to 1700 m.a.s.l. Mean annual temperature varies between 1.5 and 4°C, with mean growing season (May to October) temperature ranges of 7.5 to 10°C, and an annual precipitation of about 800 to 2000 mm. Bedrock and soils are variable, with Podzols, Cambisols and Leptosols making up the predominant soil types [32]. Norway spruce is the dominant tree species, mixed with minor components (less than 5%) of rowan (*Sorbus aucuparia* L.) and stone pine (*Pinus cembra* L.). The understory is dominated by bilberry (*Vaccinium myrtillus* L.), hairy reed grass (*Calamagrostis villosa* (Chaix) J. F. Gmelin), greater wood-rush (*Luzula sylvatica* Huds.) and wavy hair-grass (*Avenella flexuosa* (L.) Trin.). In these forests, disturbance is primarily caused by wind and the European spruce bark beetle, *Ips typographus* (L.) [21].

## (b) Forest structure and dendrochronological data

We used a hierarchical sampling framework to analyse the effect of historical disturbances on biodiversity and carbon dynamics. During the years 2011–2014, within each landscape (eastern, western, southern Carpathians) we studied 10–12 stands and established a series of 1000 m² sample plots using a stratified random design. The approximate size of the sampled landscape was roughly 10 000 km², and each stand was approximately 100 ha in size. We used a regular grid with cells of two hectares and randomly placed a circular plot within each grid cell. In total, we sampled 309 plots, representing an average of 12 plots per stand.

Within each plot, we measured the composition and structure of living and dead standing trees. We recorded the diameter of each live and dead standing tree (diameter at breast height (DBH) ≥ 10 cm) and assigned a decay class to each dead tree

[33]. The line intersect method [33] was used to measure the amount of downed dead wood. All fallen trees or branch fragments greater than or equal to 10 cm in diameter encountered along each transect were measured and identified by species and decay class, using a total transect length of 100 m per plot, split into five sub-transects of 20 m each, evenly radiating from the plot centre. We computed the volume of downed dead wood after Harmon & Sexton [33]. We collected increment cores from 25 trees selected randomly from the non-suppressed living trees with DBH ≥10 cm in each plot. Each increment core was collected 1 m above the ground and was processed for laboratory analysis.

## (c) Disturbance history

We used disturbance chronologies from a published, approximately 250 year long record of disturbance history encompassing our study plots [21]. The chronologies delineate plot-scale past disturbance occurrences with high temporal resolution and estimate the magnitude of associated events.

These chronologies were derived from analyses of temporal patterns in inter-annual tree growth. Growth variation was quantified from measurements of annual radial increment in tree core samples, which were collected from the same survey plots used in this study. Statistically anomalous tree growth variation exceeding site-specific thresholds and sustained over minimum pre-defined temporal intervals were attributed to disturbance-driven canopy openings [21]. Disturbance severity was defined in terms of the proportional area of tree canopy removed [34]. These growth surges also defined the timing of event occurrence. Years since the main disturbance were calculated as the year of data collection minus the year of maximum severity. For recently disturbed plots, where the current canopy area disturbed was larger than dendrochronologically detected maximum disturbance severities, the severity was expressed by current canopy openness. Current canopy openness was calculated as the difference between mean canopy closure of the whole dataset and current canopy closure of a given plot. Reconstructions of individual disturbance events, based on these canopy area models, were subsequently aggregated into temporal and spatial chronologies of historical disturbance [21].

## (d) Biodiversity indicator data

Because complete biodiversity inventories are usually not feasible in forest stands, most research on the relationships between biodiversity and ecosystem functioning relies on indicators, either based on forest structural features (i.e. habitat-based), or on the diversity of one or more species (i.e. taxa-based) [35]. Here, we used both approaches employing a biodiversity potential index (BPI) based on forest structure [36] and presence/absence data of a key umbrella bird species for the study region—the capercaillie [37].

### (i) Biodiversity potential

The BPI is a proxy of the suitability of a given stand to sustain biodiversity and is based on a set of five basic structural attributes: (1) standing dead trees, (2) downed logs, (3) large old trees, (4) diversity of understory vegetation, and (5) light availability at the ground floor, which are equally weighted to compose a summary index [36]. Several studies have shown these attributes to be strongly predictive of some elements of biodiversity [38]. They relate, for instance, to saproxylic beetles, wood-inhabiting fungi, lichens and mosses, understory vascular plants and light demanding species of true bugs. BPI varies between 0 and 5, with a high BPI representing a highly heterogeneous and diverse stand structure. A detailed description of the BPI calculation procedure is presented in the electronic supplementary material, figures S1 and S2.

In order to evaluate the effectiveness of BPI as a proxy for biodiversity, we selected a subset of 58 plots with available data [39] on lichens and wood-inhabiting fungi that are both considered important indicators of forest continuity and naturalness [40]. Generalized linear mixed models with gamma distributions revealed a significant positive relationship between BPI and both diversity of red-listed lichen ($z = 3.75$, $p = 0.0002$) and fungi species ($z = 1.97$, $p = 0.0494$). Using the BPI to predict the diversity of those groups on new data provided highly accurate estimates with a cross-validated mean absolute percentage prediction error of 4.8 and 16.3%, respectively. Thus, we consider the BPI a useful proxy for biodiversity in primary spruce forests in the Carpathians. For further details see the electronic supplementary material and Bače et al. [36].

### (ii) Umbrella species data

We investigated the occurrence of the umbrella species capercaillie, a species of high conservation concern in Europe [37]. A capercaillie is a ground-dwelling bird species that inhabits forest habitats characterized by open canopy (40–60%), structural heterogeneity, and rich ground vegetation [41]. Capercaillies typically inhabit primary forests in the study region [42], although habitat associations may differ in other parts of Europe. We thoroughly searched the study plots for signs of capercaillie occurrence (e.g. feathers, droppings, tracks in the snow) for 15 mins, both in summer and winter seasons (one visit per season). Only presence (at least one presence, recorded during at least one visit) and absence (no sign recorded during both seasons) data were used in the analyses [39].

## (e) Carbon data
### (i) Carbon storage (aboveground carbon stocks in tree biomass)

We calculated contemporary carbon storage in different aboveground tree biomass pools using published allometric models based on DBH [39]. Specifically, aboveground living biomass, composed of stem, branch, and foliage tissue, was calculated using species-specific equations for 14 tree species [43]. Species-specific allometric equations are shown in the electronic supplementary material, table S1. Biomass of standing dead trees was determined using models from Kublin & Breidenbach [44].

Finally, the volume of forest floor deadwood was computed according to Harmon & Sexton [33] and converted to biomass using estimates of wood density that account for decay stage [45]. We subsequently summed all biomass components and approximated total aboveground carbon storage as 50% of total biomass [46].

### (ii) Carbon sequestration

Using the tree ring dataset [21], we also estimated contemporary carbon sequestration (mean rates of change in forest carbon stocks) based on total plot-level aboveground biomass increment (AGBI) averaged over the last 10 years [47]. Based on allometric models [43] (see the electronic supplementary material, table S1), we used annual DBH increases to estimate AGBI for all living plot trees for each year in the most recent 10 year interval preceding field surveys. Inter-annual DBH increases for the 10 year window were computed from measures of annually resolved radial growth obtained from tree core samples. Radial growth rates and associated DBH variation in unsampled plot trees were approximated with data from neighbouring trees in congruent size classes. Decadal mean biomass increments of individual trees were then aggregated to produce estimates of contemporary plot-scale AGBI [48].

## (f) Data analysis

Generalized additive mixed models (GAMM, [49]) with restricted maximum likelihood were used to estimate optimal disturbance conditions that support the highest biodiversity potential, the highest probability of umbrella bird species occurrence, maximum carbon sequestration and maximum carbon storage while accounting for a hierarchical structure of sampling design (plots nested within stands). Based on detailed dendrochronological measurements [21], disturbance conditions were defined as the time since the most severe disturbance and its severity per plot. Separate models were built for biodiversity and carbon dynamics data [39]. Capercaillie occurrence was fitted by binomial GAMM with a logit link function. Characteristics of carbon stocks (total biomass carbon, living biomass carbon, dead standing biomass carbon, downed dead biomass carbon and biomass carbon increment) were fitted by GAMMs with a normal error distribution and an identity link function. The fixed effects component of the GAMMs contained thin plate regression spline smoothers for year and severity of the strongest disturbance. We set the upper limit on the smooth terms to four degrees of freedom and implemented an extra penalty to allow for shrinking the effective degrees of freedom towards zero, i.e. to perform variable selection [50]. The random effect structure involved identity of stands, while landscape-level hierarchy was not formally treated in statistical modelling owing to a low number of replicates (three landscapes only). We built the random effects part of the GAMMs sequentially, first specifying models with complex random effect structure involving factor smooths, the nonlinear counterpart to the combination of random intercepts and random slopes [51]. The models were subsequently simplified to the random intercepts and random slopes. The most parsimonious random effect structure was selected using $x^2$ tests on the differences in the restricted maximum-likelihood scores [52]. We assessed model performance using diagnostic plots and square-root transformed data on standing dead wood to meet the assumptions of normality and homogeneity of variance. Because the sampling design was spatially structured, we constructed correlograms of residuals to check for autocorrelation [53] but did not find significant spatial autocorrelation.

To assess the influence of disturbances at the stand level, the plot-level disturbance and carbon data were averaged per stand, and the capercaillie occurrences were summarized per stand. The

**Table 1.** Results of GAMMs at plot (patch) scale and GAMs at stand scale testing for the effect of time since the strongest disturbance and its severity on capercaillie occurrence, biodiversity potential and characteristics of carbon stocks in primary forests. (Effective degrees of freedom (edf), test statistics ($x^2$/F) and probabilities ($p$) are displayed along with adjusted determination coefficients ($R^2$) for each model. Results significant at $\alpha = 5\%$ are highlighted in italics.)

| scale | variable | time since maximum disturbance | | | maximum disturbance severity | | | |
|---|---|---|---|---|---|---|---|---|
| | | edf | $x^2$/F | $p$ | edf | $x^2$/F | $p$-value | $R^2$ |
| plot (patch) | capercaillie occurrence | <0.1 | <0.1 | 0.759 | *1.5* | *7.3* | *0.034* | 0.13 |
| | biodiversity potential | *3.9* | *3.7* | *<0.001* | *1.0* | *3.8* | *<0.001* | 0.27 |
| | carbon stock | *2.9* | *10.4* | *<0.001* | 1.6 | 0.7 | 0.082 | 0.37 |
| | carbon sequestration | *7.4* | *30.1* | *<0.001* | *0.8* | *0.7* | *0.038* | 0.51 |
| stand | capercaillie occurrence | <0.1 | <0.1 | 0.658 | 0.6 | 0.5 | 0.113 | 0.06 |
| | biodiversity potential | 0.8 | 0.4 | 0.248 | *1.4* | *2.4* | *0.013* | 0.24 |
| | carbon stock | 1.0 | 2.3 | 0.139 | 1 | <0.1 | 0.981 | <0.01 |
| | carbon sequestration | <0.1 | <0.1 | 0.484 | 0.4 | 0.2 | 0.289 | 0.02 |

stand-level data were fitted using generalized additive models (GAM, [49]) with the same settings as the GAMMs above. The performance of plot-level GAMMs and stand-level GAMs was compared using adjusted determination coefficients ($R^2$). The $R^2$ was defined as the proportion of variance explained, where original variance and residual variance are both estimated using unbiased estimators penalizing for number of predictors [54].

To investigate variability of forest co-benefits over multiple spatial scales, we calculated coefficients of variation of the observed values among plots (patches), stands and landscapes and plotted the estimates for each forest function.

The analyses were performed in R [55] using the libraries itsadug [52], mgcv [49] and ncf [53].

## 3. Results

Our results revealed that past disturbances had significant and century-long effects on contemporary forest functions (carbon storage, carbon sequestration, capercaillie occurrence, and structure-based biodiversity potential) (table 1 and figure 2; electronic supplementary material, figure S3).

At the plot (patch) scale, total carbon storage was highest in sites that were strongly disturbed (*ca* 30–70% canopy removed) a century or two ago. By contrast, the highest rates of carbon sequestration occurred in more recently (*ca* 50 years ago) disturbed sites that experienced a broad range of disturbance severity (optimum from *ca* 20 to 80% canopy removed) (figure 2; electronic supplementary material, figure S3). Biodiversity potential in primary forests showed a bimodal, U-shaped response on disturbance severity and was high under a broad variety of disturbance conditions. Specifically, the BPI was highest in recently disturbed forests and those disturbed two centuries ago, covering a wide range of disturbance severities (*ca* 15–75%). Finally, moderate severity disturbances (*ca* 25–40% canopy removed) increase the probability of capercaillie presence irrespective of disturbance timing (figure 2; electronic supplementary material, figure S3). Aboveground carbon storage reached maximum values under different disturbance conditions than carbon sequestration, probability of capercaillie occurrence, and biodiversity potential (figure 2; electronic supplementary material, figure S3).

At the stand level, the influence of disturbance characteristics was less pronounced, and the GAMs exhibited considerably lower explanatory power than the corresponding plot-level GAMMs (table 1). Similarly, variability of all forest functions decreased with increasing spatial scale (figure 3), demonstrating that natural disturbance regimes generate fluctuating trade-offs in ecosystem services at local scales but maintain an overall homeostasis or stability of co-benefits over large regions.

## 4. Discussion

Carbon storage and biodiversity are interrelated ecosystem functions [56], which fluctuate over time under natural disturbance regimes. Different seral conditions and variable successional pathways create a diversity of ecosystem functions. Here, while biodiversity potential had a U-shaped response to time since disturbance in these unmanaged spruce stands, being highest early after disturbance and then in later stages of forest development, carbon sequestration and stocks peaked in early-successional and old-growth stages, respectively. Recent disturbances increased light and deadwood availability, conditions known to benefit many elements of forest biodiversity [57]. As soon as the forest canopy closed, reduced light availability and more homogeneous forest structure resulted in decreased biodiversity potential. Meanwhile, carbon sequestration was highest immediately following disturbance, bolstered by the rapid growth of younger trees already in the understory owing to advanced regeneration [47]. By contrast, aboveground carbon storage was higher in old-growth forest development stages, particularly in large stems and dead wood [58,59]. Our results indicate that the later stages of forest development after disturbance again increased the biodiversity potential associated with complex forest structures (figure 2).

Interestingly, rather than timing, severity was the most important disturbance feature for capercaillies. The probability of capercaillie occurrence was high across the full range of time since disturbance, as long as the disturbance severity was moderate. This contrasts with a study from the Bavarian forest that found increasing habitat suitability with time since disturbance and a positive effect of high severity disturbances [60]. The contrasting results might depend on regional differences that influence successional

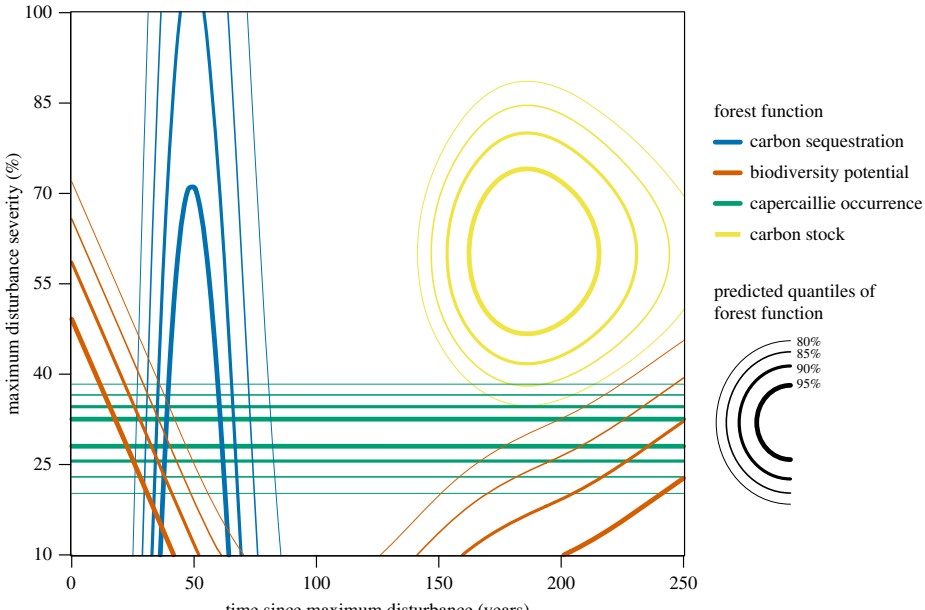

**Figure 2.** Maxima of forest functions along the gradients of maximum disturbance severity and time since that event. Isolines represent upper percentiles (greater than 80%) of GAMM-predicted values of the forest functions.

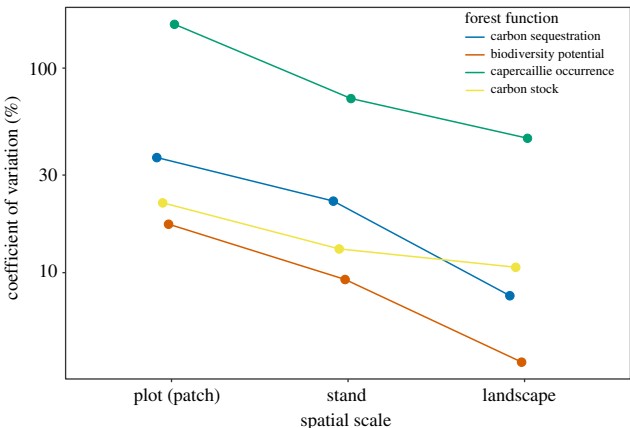

**Figure 3.** Coefficient of variation of forest functions calculated among plots (patches), stands and landscapes. Note that the ordinate is logarithmically scaled.

pathways, or they may be attributable to Kortmann *et al.* [60] only investigating dynamics over two decades following disturbance, a limited timeframe that could correspond to short-term post-disturbance reproductive success. The preference of capercaillies to moderately disturbed plots independently of disturbance timing as found here may also be explained by the fact that moderate severity disturbances—both recent and further in the past—generally lead to high structural complexity in the studied primary forests [26]. In general, moderate severity disturbance could result in an optimal balance, under which several forest functions can reach relatively high levels (see the overlaps in figure 2).

Whether forests and forest management for carbon storage can jointly achieve climate change mitigation goals and sufficient quantity and quality habitat for rare species and biodiversity is a key topic in conservation research and policy [10]. Our results highlight the importance of spatial and temporal scales when accounting for relationships between forest biodiversity and carbon functions [61]. Although it may prove challenging to simultaneously maximize total carbon storage, sequestration and biodiversity maintenance at small spatial scales, our results show that natural disturbance regimes can maintain relatively high levels of all functions in Carpathian spruce-dominated landscapes not subject to forest management [62].

As such, primary forests represent both important carbon stores and biodiversity hotspots [29]. Although optimal site conditions for carbon and biodiversity may be associated with different disturbance histories, it is important to highlight the positive relationship between carbon stocks and biodiversity potential and the lack of a significant difference between carbon stocks on plots with and without capercaillie occurrence (electronic supplementary material, figure S4). In general, carbon storage values are significantly higher in primary forests compared to mature managed forests under the same site conditions (elevation, soil etc.) [63]. Thus, despite the local fluctuation caused by natural disturbance, our study supports the conservation of unmanaged forest landscapes as an effective tool to promote both biodiversity and carbon co-benefits.

## 5. Study caveats

While our statistical model and covariation analyses relied on established approaches, our study has some limitations that warrant discussion. First, it only focused on aboveground tree carbon without considering soil, which can form a considerable proportion of total forest carbon [56]. An intensive soil sampling of the study area would address this issue but was beyond the scope and capacity of the current study [62]. Therefore, the peak in carbon values 200 years after natural disturbance should be interpreted with caution because including soil data could show longer-term increases in total carbon. Second, we used two biodiversity indicators as a proxy of biodiversity. While we concede that a multi-taxon approach could provide more appropriate results, the biodiversity potential index was a reliable predictor of species richness of wood-inhabiting fungi and lichens (see §2d(i);

electronic supplementary material). Moreover, capercaillies have been widely used in conservation planning and shown to be a suitable umbrella species for rare forest bird species occurrence [37]. Thus, we believe that our analyses produced results with a high degree of generality.

# 6. Conclusion

Our results significantly enhance our understanding of the effects of historical disturbance on contemporary ecosystem co-benefits and trade-offs. In particular, they emphasize that accounting for long-term variation of past disturbance could improve current policies aimed at mitigating climate change and biodiversity loss. Disturbances have long-lasting effects on forest functions and post-disturbance successional pathways [26]. Clearly, accounting for long time scales and alternative post-disturbance development trajectories poses a significant challenge to designing effective conservation and mitigation strategies, particularly given projected changes in disturbance regimes. Our results suggest that these challenges can be addressed by embracing a landscape perspective. While carbon sequestration and storage or biodiversity cannot be maximized everywhere on small spatial scales, a larger landscape has the capacity to deliver optimal levels of biodiversity and carbon co-benefits. A variety of disturbance spatial scales and temporal frequencies are needed to foster both carbon sequestration and stocks, and to maintain high levels of biodiversity. Because all three objectives cannot be simultaneously maximized in small reserves, it is important to delineate large tracts of strictly protected forest landscapes to maintain a range of seral stages under a regime of natural disturbances. The size of such protected areas could be guided by the minimum dynamic area framework, which would help to determine the minimum reserve size required to incorporate natural disturbance regimes and maintain ecological processes [64]. Furthermore, forests must be allowed to attain older ages if they are to reach their biodiversity and carbon storage potential [65,66]. Thus, protecting existing primary forests and increasing the size of strictly protected forest landscapes (e.g. rewilding) is necessary to encompass shifting patch mosaics driven by a wide range of disturbances. These strategies would help maintain a range of ecosystem functions in times of accelerating environmental change.

Data accessibility. Data are available from the Dryad Digital Repository: https://doi.org/10.5061/dryad.0k6djhb13 [39].

Authors' contributions. M.M.: conceptualization, data curation, investigation, methodology, writing—original draft; M.Svi.: formal analysis, investigation, writing—review and editing; R.B.: data curation, methodology, writing—review and editing; W.S.K.: conceptualization, writing—review and editing; V.T.: conceptualization, writing—review and editing; G.W.M., H.K., A.B., D.K., K.Bol., K.Beg., V.C., O.C., D.R., M.D., M.Fer., M.Fra., R.G., J.H., P.J., O.K., J.L., L.M., T.N., J.P., J.L.P., R.R., C.C.R., M.R., F.S., J.S., M.Syn., O.V., V.Z.: writing—review and editing; M.Svo.: funding acquisition, resources, supervision. All authors gave final approval for publication and agreed to be held accountable for the work performed therein.

Competing interests. We declare we have no competing interests.

Funding. Funding for this research was provided by the Czech Science Foundation (grant GACR no. 21–27454S) and the institutional project no. CZ.02.1.01/0.0/0.0/16_019/0000803. M.S. was supported by the Operational Programme Integrated Infrastructure (OPII) funded by the ERDF (ITMS313011T721).

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
