## [Peer Review File · Proceedings of the Royal Society B: Biological Sciences]

Review History

RSPB-2021-1631.R0 (Original submission)

Review form: Reviewer 1

Recommendation

Accept with minor revision (please list in comments)

Scientific importance: Is the manuscript an original and important contribution to its field?

Good

General interest: Is the paper of sufficient general interest?

Good

Quality of the paper: Is the overall quality of the paper suitable?

Excellent

Is the length of the paper justified?

Yes

Should the paper be seen by a specialist statistical reviewer?

No

Do you have any concerns about statistical analyses in this paper? If so, please specify them explicitly in your report.

No

It is a condition of publication that authors make their supporting data, code and materials available - either as supplementary material or hosted in an external repository. Please rate, if applicable, the supporting data on the following criteria.

Is it accessible?

N/A

Is it clear?

N/A

Is it adequate?

N/A

Do you have any ethical concerns with this paper?

No

Comments to the Author

This is a very well executed study and well written report. I have only four strong requests for clarification or expansion:

- 1) It is not clear to me what role the collection of new dendrochronological data was: only for growth measurements (referred to line 257), or to determine time since disturbance? (lines 269-270). The respective roles of these new data and the disturbance histories compiled from the Schurman et al. 2018 data set need to be more clearly outlined.
- 2) It should be made clear that the modelling approach was strictly a statistical approach, not a simulation approach: your abstract (line 42), in which you state "...we modelled the effect of 250 years of disturbances..." leads the reader to expect some sort of stochastic simulation of disturbance severity and frequency.
- 3) The resulting response patterns (U-shaped for biodiversity, rising for C storage, early peak then declining for growth or C sequestration) have been observed or postulated in the North American literature since the 1980s. I don't have access to my library just now, but I think there may have been a figure as early as 1981 in the Shugart & West book on forest succession. See articles by Trofymow et al. 2003 (Environmental Reviews 11:S187-S204), Franklin et al. 2002 (For.Ecol.Manage. 155: 399-423), and Harmon & Pabst 2015 (J.Veg.Sci. 26:722-732) and articles cited therein, including Spies & Franklin 1988 and Spies 1998.
- 4) Please expand your discussion or add some conclusions regarding the conservation and management implications: that a variety of disturbance scales and frequencies are needed to foster both C sequestration and stocks? That all objectives cannot be met in small reserves of primary forest, but can be if a sufficiently large area or number of reserves is protected? That managed forests must be allowed to attain great age if they are to reach their biodiversity and C storage potential?

My only other suggestions are minor editorial ones:

L. 45: short term [no hyphen in this case]

Lines 121 and 124: please provide taxonomic authorities for *Tatrao urogallus* and *Picea abies*

L. 142: likewise, for consistency, please provide a common name (hairy reedgrass?) and the taxonomic authority for *Calamagrostis villosa*

- L. 145: taxonomic authority needed for *Ips typographus*
- L. 175: rephrase as simply "above the ground" [delete "level surface"]
- L. 232: "that" instead of "which"
- L. 234: "inhabit" instead of "inhabits"
- L. 268-270: as noted above, do these "detailed dendrochronological measurements" refer to the new tree ring data collected, or to eh Schurman et al (2018) data set?
- L. 281: comma, not semi-colon
- L. 288: "Because" instead of "Since"
- L. 291-292: rephrase without parentheses "... plot-level data to generate values at the stand level..."
- L. 299: what reference for R package mgcv? [not "70"]
- I found no such minor errors in the Results and Discussion – nicely written.

Review form: Reviewer 2

Recommendation

Major revision is needed (please make suggestions in comments)

Scientific importance: Is the manuscript an original and important contribution to its field?

Good

General interest: Is the paper of sufficient general interest?

Good

Quality of the paper: Is the overall quality of the paper suitable?

Good

Is the length of the paper justified?

Yes

Should the paper be seen by a specialist statistical reviewer?

No

Do you have any concerns about statistical analyses in this paper? If so, please specify them explicitly in your report.

No

It is a condition of publication that authors make their supporting data, code and materials available - either as supplementary material or hosted in an external repository. Please rate, if applicable, the supporting data on the following criteria.

Is it accessible?

Yes

Is it clear?

Yes

Is it adequate?

Yes

Do you have any ethical concerns with this paper?

No

Comments to the Author

The manuscript presents a study carried out in Primary Forest in the Carpathian Mountain ecoregion, which encompasses the majority of spruce forests in Europe. The design is balanced, and there are thirty primary forest stands. Although some information lacks in the methods for a complete understanding of the procedures done, data collection is appropriate. The analyses employed are also appropriate. The results are interesting and clearly show that legacies of disturbance over centuries have shaped contemporary forestry. Overall, the organization of the manuscript is satisfactory, and it has the potential to be published. However, some points need to be revised.

Objectives and Introduction

The questions stated are adequate given the subject. Moreover, the introduction is well-written and appropriately structured. The specific comment is:

Line 126-130: The questions have been dealt with in a general way. I believe that if they are put in a specific way, it will provide specific insight into how they were addressed.

Materials and Methods

The description of materials and methods is sufficiently informative to allow replication of the procedures. Specific comments are as follows:

Line 134-145: Please, provide a clear description of the climate (especially seasonality and temperature), topography, soil range and elevation

Line 139: "Thirty primary forest stands with no signs of human management...". It was unclear what was considered signs of human management. I believe the authors have tried to eliminate some words, but this issue needs to be clarified.

Line 159: Add the sampling period.

Line 188-198: Add the allometric equation.

Line 241: Add the allometric equations.

Line 250: Add reference.

Line 251: Add the allometric equations.

Line 296-299: Something has failed here.

Results

In general, the results are clearly represented, and all figures and tables are necessary to understand the results. Specific comments:

Line 323: You need to provide how the coefficient of determination was calculated (R^2) and equation for regression analysis. The same comment is valid for the figures;

Line 304: At this moment, it is clear that you need to add the scale of work into your work issues.

Discussion and conclusion.

Some issues were missed in the discussion:

Line 351: Do these old-growth forests refer to your primary forests?

Line 352-354: From your results, provide a direct message on what they indicate.

Line 382: Yes, it seems important to me. But not at this point in the manuscript. Provide it as a

new question in your manuscript. If you do not agree, assess the possibility of additional supplementary material.

Line 395: Please provide a new section: Study caveats

References

The references are adequate.

Decision letter (RSPB-2021-1631.R0)

27-Aug-2021

Dear Dr Mikoláš:

Your manuscript has now been peer reviewed and the reviews have been assessed by an Associate Editor. The reviewers' comments (not including confidential comments to the Editor) and the comments from the Associate Editor are included at the end of this email for your reference. As you will see, the reviewers and the Editors have raised some concerns with your manuscript and we would like to invite you to revise your manuscript to address them.

Research ethics:

Use of animals and field studies:

If your study uses animals please include details in the methods section of any approval and licences given to carry out the study and include full details of how animal welfare standards were ensured. Field studies should be conducted in accordance with local legislation; please

include details of the appropriate permission and licences that you obtained to carry out the field work.

It is a condition of publication that you make available the data and research materials supporting the results in the article. Please see our Data Sharing Policies (<https://royalsociety.org/journals/authors/author-guidelines/#data>). Datasets should be deposited in an appropriate publicly available repository and details of the associated accession number, link or DOI to the datasets must be included in the Data Accessibility section of the article (<https://royalsociety.org/journals/ethics-policies/data-sharing-mining/>). Reference(s) to datasets should also be included in the reference list of the article with DOIs (where available).

If you wish to submit your data to Dryad (<http://datadryad.org/>) and have not already done so you can submit your data via this link [http://datadryad.org/submit?journalID=RSPB&manu=\(Document not available\)](http://datadryad.org/submit?journalID=RSPB&manu=(Document%20not%20available)), which will take you to your unique entry in the Dryad repository.

Please submit a copy of your revised paper within three weeks. If we do not hear from you within this time your manuscript will be rejected. If you are unable to meet this deadline please let us know as soon as possible, as we may be able to grant a short extension.

Best wishes,
Dr Sasha Dall
mailto: proceedingsb@royalsociety.org

Associate Editor
Board Member: 1
Comments to Author:

Thank you for your submission to the Proceedings of the Royal Society. Two reviewers have evaluated your paper and both have some suggestions for improvement. Please revise that paper in a way that considers their comments. I particularly stress the suggestion of review 1 that you

enhance the conservation implications. Reviewer 2 is asking for some additional methodological details and a discussion of caveats. We look forward to receiving a revised version of this manuscript.

Reviewer(s)' Comments to Author:

Referee: 1

Comments to the Author(s)

This is a very well executed study and well written report. I have only four strong requests for clarification or expansion:

- 1) It is not clear to me what role the collection of new dendrochronological data was: only for growth measurements (referred to line 257), or to determine time since disturbance? (lines 269-270). The respective roles of these new data and the disturbance histories compiled from the Schurman et al. 2018 data set need to be more clearly outlined.
- 2) It should be made clear that the modelling approach was strictly a statistical approach, not a simulation approach: your abstract (line 42), in which you state "...we modelled the effect of 250 years of disturbances..." leads the reader to expect some sort of stochastic simulation of disturbance severity and frequency.
- 3) The resulting response patterns (U-shaped for biodiversity, rising for C storage, early peak then declining for growth or C sequestration) have been observed or postulated in the North American literature since the 1980s. I don't have access to my library just now, but I think there may have been a figure as early as 1981 in the Shugart & West book on forest succession. See articles by Trofymow et al. 2003 (Environmental Reviews 11:S187-S204), Franklin et al. 2002 (For.Ecol.Manage. 155: 399-423), and Harmon & Pabst 2015 (J.Veg.Sci. 26:722-732) and articles cited therein, including Spies & Franklin 1988 and Spies 1998.
- 4) Please expand your discussion or add some conclusions regarding the conservation and management implications: that a variety of disturbance scales and frequencies are needed to foster both C sequestration and stocks? That all objectives cannot be met in small reserves of primary forest, but can be if a sufficiently large area or number of reserves is protected? That managed forests must be allowed to attain great age if they are to reach their biodiversity and C storage potential?

My only other suggestions are minor editorial ones:

L. 45: short term [no hyphen in this case]

Lines 121 and 124: please provide taxonomic authorities for *Tatrao urogallus* and *Picea abies*

L. 142: likewise, for consistency, please provide a common name (hairy reedgrass?) and the taxonomic authority for *Calamagrostis villosa*

L. 145: taxonomic authority needed for *Ips typographus*

L. 175: rephrase as simply "above the ground" [delete "level surface"]

L. 232: "that" instead of "which"

L. 234: "inhabit" instead of "inhabits"

L. 268-270: as noted above, do these "detailed dendrochronological measurements" refer to the new tree ring data collected, or to eh Schurman et al (2018) data set?

L. 281: comma, not semi-colon

L. 288: "Because" instead of "Since"

L. 291-292: rephrase without parentheses "... plot-level data to generate values at the stand level..."

L. 299: what reference for R package *mgcv*? [not "70"]

I found no such minor errors in the Results and Discussion - nicely written.

Referee: 2

Comments to the Author(s)

The manuscript presents a study carried out in Primary Forest in the Carpathian Mountain ecoregion, which encompasses the majority of spruce forests in Europe. The design is balanced, and there are thirty primary forest stands. Although some information lacks in the methods for a complete understanding of the procedures done, data collection is appropriate. The analyses employed are also appropriate. The results are interesting and clearly show that legacies of

disturbance over centuries have shaped contemporary forestry. Overall, the organization of the manuscript is satisfactory, and it has the potential to be published. However, some points need to be revised.

Objectives and Introduction

The questions stated are adequate given the subject. Moreover, the introduction is well-written and appropriately structured. The specific comment is:

Line 126-130: The questions have been dealt with in a general way. I believe that if they are put in a specific way, it will provide specific insight into how they were addressed.

Materials and Methods

The description of materials and methods is sufficiently informative to allow replication of the procedures. Specific comments are as follows:

Line 134-145: Please, provide a clear description of the climate (especially seasonality and temperature), topography, soil range and elevation

Line 139: "Thirty primary forest stands with no signs of human management...". It was unclear what was considered signs of human management. I believe the authors have tried to eliminate some words, but this issue needs to be clarified.

Line 159: Add the sampling period.

Line 188-198: Add the allometric equation.

Line 241: Add the allometric equations.

Line 250: Add reference.

Line 251: Add the allometric equations.

Line 296-299: Something has failed here.

Results

In general, the results are clearly represented, and all figures and tables are necessary to understand the results. Specific comments:

Line 323: You need to provide how the coefficient of determination was calculated (R^2) and equation for regression analysis. The same comment is valid for the figures;

Line 304: At this moment, it is clear that you need to add the scale of work into your work issues.

Discussion and conclusion.

Some issues were missed in the discussion:

Line 351: Do these old-growth forests refer to your primary forests?

Line 352-354: From your results, provide a direct message on what they indicate.

Line 382: Yes, it seems important to me. But not at this point in the manuscript. Provide it as a new question in your manuscript. If you do not agree, assess the possibility of additional supplementary material.

Line 395: Please provide a new section: Study caveats

References

The references are adequate.

Author's Response to Decision Letter for (RSPB-2021-1631.R0)

See Appendix A.

Decision letter (RSPB-2021-1631.R1)

27-Sep-2021

Dear Dr Mikoláš

I am pleased to inform you that your manuscript entitled "Natural disturbance impacts on trade-offs and co-benefits of forest biodiversity and carbon" has been accepted for publication in Proceedings B.

Data Accessibility section

Open Access

Paper charges

Sincerely,

Dr Sasha Dall

Appendix A

Associate Editor

Board Member: 1

Comments to Author:

Thank you for your submission to the Proceedings of the Royal Society. Two reviewers have evaluated your paper and both have some suggestions for improvement. Please revise that paper in a way that considers their comments. I particularly stress the suggestion of review 1 that you enhance the conservation implications. Reviewer 2 is asking for some additional methodological details and a discussion of caveats. We look forward to receiving a revised version of this manuscript.

Thank you very much for considering our manuscript. In the revised manuscript, we addressed all the comments of both reviewers. We enhanced the conservation implications, added the requested methodological details, created a separate section on discussion of caveats, and moved some information to the supplement as suggested. We carefully considered and addressed all the proposed changes throughout the manuscript.

Reviewer(s)' Comments to Author:

Referee: 1

Comments to the Author(s)

This is a very well executed study and well written report.

Thank you very much for the positive evaluation of our manuscript and valuable review that helped to improve our study.

I have only four strong requests for clarification or expansion:

1) It is not clear to me what role the collection of new dendrochronological data was: only for growth measurements (referred to line 257), or to determine time since disturbance? (lines 269-270). The respective roles of these new data and the disturbance histories compiled from the Schurman et al. 2018 data set need to be more clearly outlined.

We now provide additional clarification in the revised method section. We did not collect any new tree ring data. All the dendrochronological data used in this study are published in Schurman et al. 2018.

2) It should be made clear that the modelling approach was strictly a statistical approach, not a simulation approach: your abstract (line 42), in which you state "...we modelled the effect of 250 years of disturbances..." leads the reader to expect some sort of stochastic simulation of disturbance severity and frequency.

Corrected. Now the sentence in the abstract states:

"Using a tree-ring-based approach, we analysed the effect of 250 years of disturbances on present-day biodiversity indicators and carbon dynamics in primary forests."

3) The resulting response patterns (U-shaped for biodiversity, rising for C storage, early

peak then declining for growth or C sequestration) have been observed or postulated in the North American literature since the 1980s. I don't have access to my library just now, but I think there may have been a figure as early as 1981 in the Shugart & West book on forest succession. See articles by Trofymow et al. 2003 (Environmental Reviews 11:S187-S204), Franklin et al. 2002 (For.Ecol.Manage. 155: 399-423), and Harmon & Pabst 2015 (J.Veg.Sci. 26:722-732) and articles cited therein, including Spies & Franklin 1988 and Spies 1998.

Thank you very much for the suggested studies. We refer to them in the revised introduction and discussion section.

4) Please expand your discussion or add some conclusions regarding the conservation and management implications: that a variety of disturbance scales and frequencies are needed to foster both C sequestration and stocks? That all objectives cannot be met in small reserves of primary forest, but can be if a sufficiently large area or number of reserves is protected? That managed forests must be allowed to attain great age if they are to reach their biodiversity and C storage potential?

Thank you very much for this suggestion. We expanded the discussion and provided conclusions regarding the conservation and management implications in more detail.

Lines 381-395:

“While carbon sequestration and storage or biodiversity cannot be maximized everywhere on small spatial scales, a larger landscape has the capacity to deliver optimal levels of biodiversity and carbon co-benefits. A variety of disturbance spatial scales and temporal frequencies are needed to foster both C sequestration and stocks, and to maintain high levels of biodiversity. Because all three objectives cannot be simultaneously maximized in small reserves, it is important to delineate large tracts of strictly protected forest landscapes to maintain a range of seral stages under a regime of natural disturbances. The size of such protected areas could be guided by the minimum dynamic area framework, which would help to determine the minimum reserve size required to incorporate natural disturbance regimes and maintain ecological processes [64]. Furthermore, forests must be allowed to attain older ages if they are to reach their biodiversity and C storage potential [65,66]. Thus, protecting existing primary forests and increasing the size of strictly protected forest landscapes (e.g. rewilding) is necessary to encompass shifting patch mosaics driven by a wide range of disturbances. These strategies would help maintain a range of ecosystem functions in times of accelerating environmental change.”

My only other suggestions are minor editorial ones:

L. 45: short term [no hyphen in this case]

Corrected.

Lines 121 and 124: please provide taxonomic authorities for Tatrao urogallus and Picea abies

Done.

L. 142: likewise, for consistency, please provide a common name (hairy reedgrass?) and the taxonomic authority for Calamagrostis villosa

Done.

L. 145: taxonomic authority needed for Ips typographus

Done.

L. 175: rephrase as simply “above the ground” [delete “level surface”]

Done.

L. 232: “that” instead of “which”

Done.

L. 234: “inhabit” instead of “inhabits”

Corrected.

L. 268-270: as noted above, do these “detailed dendrochronological measurements” refer to the new tree ring data collected, or to eh Schurman et al (2018) data set?

All dendrochronological measurements are from Schurman et al. 2018. We added a reference in the text and hope that this section will be clear for the readers.

L. 281: comma, not semi-colon

Corrected.

L. 288: “Because” instead of “Since”

Corrected.

L. 291-292: rephrase without parentheses “... plot-level data to generate values at the stand level...”

Rephrased as suggested by the reviewer.

L. 299: what reference for R package mgcv? [not “70”]

Corrected.

I found no such minor errors in the Results and Discussion – nicely written.

Thank you very much.

Referee: 2

Comments to the Author(s)

The manuscript presents a study carried out in Primary Forest in the Carpathian Mountain ecoregion, which encompasses the majority of spruce forests in Europe. The design is balanced, and there are thirty primary forest stands. Although some information lacks in the methods for a complete understanding of the procedures done, data collection is appropriate. The analyses employed are also appropriate. The results are interesting and clearly show that legacies of disturbance over centuries have shaped contemporary forestry. Overall, the organization of the manuscript is satisfactory, and it has the potential to be published. However, some points need to be revised.

Thank you very much for the positive evaluation of our manuscript.

Objectives and Introduction

The questions stated are adequate given the subject. Moreover, the introduction is well-written and appropriately structured. The specific comment is:

Line 126-130: The questions have been dealt with in a general way. I believe that if they are put in a specific way, it will provide specific insight into how they were addressed.

We have revised the questions to make them more specific. We now also explain in greater detail our approach in the paragraph before the questions, where we provide more insight into the methods we apply.

The entire section reads as follows:

*“Here, we investigated the long-term response of biodiversity indicators (biodiversity potential index and occurrence of an umbrella species, the capercaillie (*Tetrao urogallus* L.), and forest carbon dynamics (sequestration and total storage) to 250 years of disturbance history across a gradient of disturbance severity and timing. To reconstruct disturbance histories, we collected 7,725 tree cores in 30 of the best-preserved primary Norway spruce (*Picea abies* (L.) Karst.) forest stands in temperate Europe. We addressed three main research questions:*

(1) How does variation in past disturbance history affect contemporary patterns of biodiversity indicators, carbon storage, and carbon sequestration? (2) What is the relative importance of disturbance severity and timing in determining contemporary biodiversity indicators and carbon storage and sequestration? (3) Under which disturbance conditions are there co-benefits vs. trade-offs between forest biodiversity, carbon storage and sequestration?”

Materials and Methods

The description of materials and methods is sufficiently informative to allow replication of the procedures.

Thank you very much for your positive evaluation.

Specific comments are as follows:

Line 134-145: Please, provide a clear description of the climate (especially seasonality and temperature), topography, soil range and elevation.

Done.

Lines 131-135: "The studied forests occupy altitudes ranging from 1,150 to 1,700 m a.s.l. Mean annual temperature varies between 1.5 and 4 °C, with mean growing season (May to October) temperature ranges of 7.5 to 10 °C, and an annual precipitation of about 800 to 2,000 mm. Bedrock and soils are variable, with Podzols, Cambisols, and Leptosols making up the predominant soil types [32]."

Line 139: "Thirty primary forest stands with no signs of human management...". It was unclear what was considered signs of human management. I believe the authors have tried to eliminate some words, but this issue needs to be clarified.

We added a more detailed explanation and a reference with description of the methods of primary forest identification that we applied to select the studied primary forests.

Lines 129-131: "Stands with no evidence of direct human influence, such as logging or livestock grazing, were selected with the help of local experts or primary forest inventories [Mikoláš et al. 2019]."

Reference: Mikoláš, M., Ujházy, K., Jasík, M., Wiezik, M., Gallay, I., Polák, P., ... & Keeton, W. S. (2019). Primary forest distribution and representation in a Central European landscape: Results of a large-scale field-based census. *Forest Ecology and Management*, 449, 117466.

Line 159: Add the sampling period.

Done.

Line 188-198: Add the allometric equation.

We added all allometric equations to the electronic supplementary material (Table S1.).

Line 241: Add the allometric equations.

We added all allometric equations to the electronic supplementary material (Table S1.).

Line 250: Add reference.

Done.

Line 251: Add the allometric equations.

We added all allometric equations to the electronic supplementary material (Table S1.).

Line 296-299: Something has failed here.

To clarify the intended meaning, this was rephrased as follows: “To investigate variability of forest co-benefits over multiple spatial scales, we calculated coefficients of variation of the observed values among plots (patches), stands, and landscapes and plotted the estimates for each forest function.”

Results

In general, the results are clearly represented, and all figures and table are necessary to understand the results. Specific comments:

Line 323: You need to provide how the coefficient of determination was calculated (R^2) and equation for regression analysis. The same comment is valid for the figures;

The adjusted R^2 was calculated according to Wherry’s formula as implemented in R. The information was added in the M&M section [l. 274-276].

Specifying a regression equation for a GA(M)M would not be very informative and difficult to sum-up algebraically since the approach uses thin plate spline smoothers which are sums of so-called basis functions where each bears its own coefficients. For example, the GAMM for capercaillie occurrence would have an equation with 65 coefficients, which would be difficult (or even impossible) to grasp in their numeric form. Therefore, the results of GA(M)Ms are usually presented graphically with summary statistics as we have done.

Line 304: At this moment, it is clear that You need to add the scale of work into your work issues.

We described details of the scale of the work in the section 2.2. Forest structure and dendrochronological data. Now it states:

Lines 144-148: “During the years 2011-2014, within each landscape (Eastern, Western, Southern Carpathians) we studied 10-12 stands and established a series of 1000 m² sample plots using a stratified random design. The approximate size of the sampled landscape was roughly 10 000 km², and each stand was approximately 100 ha in size.”

Discussion and conclusion.

Some issues were missed in the discussion:

Line 351: Do these old-growth forests refer to your primary forests?

We are now more specific. We understand the old growth stage as one of the stages within primary forest mosaics (just as the early seral stage can also be present in primary forests). Now the sentence states:

“Aboveground carbon storage was higher in old-growth forest development stages with more diverse reservoirs of stored C [Magnani et al. 2007].“

Line 352-354: From your results, provide a direct message on what they indicate.

Done.

Line 382: Yes, it seems important to me. But not at this point in the manuscript. Provide it as a new question in your manuscript. If you do not agree, assess the possibility of additional supplementary material.

We agree and we we now include the Figure in electronic supplementary material (Fig. S4) and provide more information on the relationship between carbon stock and biodiversity indicators.

Line 395: Please provide a new section: Study caveats

Done.